# The Impact of an Environmental Way of Customer's Thinking on a Range of Choice from Transport Routes in Maritime Transport

Andrej David [1,*], Peter Mako [1], Jan Lizbetin [2] and Patrik Bohm [3]

1 Department of Water Transport, Faculty of Operation and Economics of Transport and Communications, University of Zilina, Univerzitná 8215/1, 010 26 Žilina, Slovakia; peter.mako@stud.uniza.sk

2 Department of Transport and Logistics, Faculty of Technology, Institute of Technology and Business in České Budějovice, Okružní 517/10, 370 01 České Budějovice, Czech Republic; lizbetin@mail.vstecb.cz

3 Department of Quantitative Methods and Economic Informatics, Faculty of Operation and Economics of Transport and Communications, University of Zilina, Univerzitná 8215/1, 010 26 Žilina, Slovakia; patrik.bohm@fpedas.uniza.sk

* Correspondence: andrej.david@fpedas.uniza.sk

**Abstract:** The paper deals with the impact that an environmental way of thinking has on shipping and transport company customers regarding their preferences in choosing a transport route. Nowadays, maritime transport plays a very important role mainly in transoceanic container transport. It also deals with the statistics focused on container shipping, especially between North America and Europe. These statistics contribute to a general description of the development of container shipping on the route that is applicated in this case study. The significant impact of this kind of transport also reflects the estimation of the future development of container transport on the selected transport route. In this view, the least square method is used in this paper. This method can present the trend of development according to statistics. Thanks to these materials, this paper estimates a slight increase of the number of containers transported between North America and Europe in the near future. This increase will have a certain effect on the environment. Thus, as part of their business policy of sustainability and environment protection, customers will prefer a mode of transport and transport routes featuring a smaller effect on the environment in the future. The relevance of such a change in preferences in planning transport routes for the customer is reflected in the case study presented in this paper. So, one part of this paper is also dedicated to information about the impact of maritime transport on the environment. This part also explains the impact according to different studies that have been published in the last few years. The main contribution of this paper is also to point out the importance of this factor for the preferences of customers via the multi-criteria decision method. Using a multi-criteria decision method, it outlines how the factor of the impact on the environment can significantly change the offer made by a transport or shipping company, and thus how it represents a key element of whether the customer would prefer the given offer or focus on a competitor's offer.

**Keywords:** environmental; maritime transport; transport routes; AHP method

## 1. Introduction

In container transport, many companies attempt to orientate towards the most profitable offer depending on the wishes and needs of a particular customer. The evaluation of offers often starts with looking at different variants with their own indicators; many times, they play a key role in making decisions to determine a recommended proposal for a transport route with regard to customers' requirements. The environmental character of transport can represent one of these aspects for the customer. This aspect is mostly considered by customers oriented towards a sustainable way of business making from the

environment protection perspective. The sustainability and the effect on the environment can have a significant impact on the decision of whether the customer would prefer the given kind of transport and the offered route [1]. Using scientific methods such as analytic hierarchy process helps freight forwarders and shipping companies promote proposals according to each customer's request. This scientific method uses the evaluation of different criteria related to transport, such as time of transport, price, possible risks, and so on. Thanks to this method, it is possible to change these criteria and to add new criteria according to each customer's request. This paper suggests adding new criteria related to the environment.

The issue of the application of the sustainability and environment protection policy is currently solved mainly from the sellers' and buyers' point of view in direct selling. Many companies do currently apply it only from the perspective of packaging and consumer behaviour during the purchase and sales of products. However, another important aspect that is not too visible for a consumer is the transport of goods from the place of their production to the place of sales. This part is less important nowadays, and all around the world, there still remains a focus on the price of transport rather than on its ecological aspect. Step by step, in the area of transport, mostly sellers have started to pay attention to various alternatives and the potential to utilise an environmental way of transport as part of the global solution of logistics services. The ecological solutions in the transport sector are currently supported by the governments and international institutions. The prioritisation of ecological forms of transport is progressively reflected on advantageous conditions provided by road, railway, as well as port operators. In the future, this support may become a key factor which, after all, will also impact costs associated with transport. Due to this, many operators of shipping lines have already been reacting to these changes. Providing a more ecological cargo transport, they create a space for the future little by little. Nowadays, more and more shipping operators use different ways of environmental protection policy. These ways could be considered into the weight of environmental criteria in the analytic hierarchy process. These policies are mainly focused on fossil fuels and greenhouse gases, but the latest trends are also focused on the manipulation with ballast water or noise control. However, today, only five main companies actively use noise control systems and renewable energy. Most of them focused their attention on greenhouse gases and fuels without sulphur content. However, we have to also admit that lots of shipping operators accept new technologies and new regulations not due to environmental protection but due to international regulations.

To study the impact of customers' sustainability and environmental thinking, it is necessary to proceed in several fundamental steps. First of all, the basic method for assessing individual alternatives must be determined; here, the possibility to assess individual factors among each other must be taken into account. Then, it is advantageous to assess individual alternatives among each other firstly without consideration of the ecological aspect of transport, and afterwards with consideration of the aspect. After all, it is appropriate to outline a change that may happen and to point out its significance from the perspective of alternatives being offered [2,3].

The comparison of results of the overall ranking of proposals without and with consideration of the ecological aspect of transport represents the aim of this paper. In other words, the paper looks at how adding the environmental aspect may change the resulting determination of the recommended route per the customer's wishes. This paper points out the importance of individual requirements of the customer in connection with the application of the sustainability and environment protection policy while determining a suitable offer of a transport route. This way, the paper attempts to clarify how these aspects can impact the customer's decision to accept or reject the given offer [4]. As the case study shows, the environmental criteria could have a significant impact on the customer's decision. However, the final decision of the customer is closely related to the weight of the environmental factor in the analytic hierarchy process. The weight of this criteria creates a significant impact on the customer's decision. So, there are lots of possibilities regarding

how to consider this factor into final recommendations for customers. Thanks to different weights in the analytic hierarchy process, it is possible to adjust these criteria according to the activity of the companies' behavior related to the environment.

From this perspective, it is advantageous to establish a procedure that could be used for the evaluation of particular transport alternatives per individual wishes of customers. One of advantageous ways is to utilise multi-criteria methods. Their application to particular cases makes it possible to meet key requirements of the customer; currently, the sustainability and impact on the environment come to the fore noticeably [5].

## 2. Literature Review

The question of the impact of maritime transport on the environment is actively dealt with in many studies, literature, and scientific papers. Tang and Gekara consistently analyse the impact of the sustainability and new business policies associated with social responsibility on the business in maritime transport. For example, they state that through the application of the sustainability and environment protection policy, the companies operating the transport by sea may positively stand out from their competition. They also claim that in the past, maritime transport became a prototype of hard capitalism under conditions of a free market that led to the exploitation of a cheap labour force and to the reduction of costs of repairs and maintenance. After a series of serious accidents, many measures have been adopted there, and today, maritime transport is considered one of the most strictly controlled industries [6].

In general, a majority of authors deal with the impact of maritime transport on the environment only. The most frequently discussed topic is the effect of emissions. James J. Corbett and Horst W. Koehler analyse a study where average values of sulphur oxides emissions emitted into the air represent 10 g/kWh, in case of nitrogen oxides emissions, the values represent 16–17 g/kWh, and in case of carbonic oxides emissions, the values represent 655–700 g/kWh [7].

Jiang J., Aksoyoglu S., Ciarelli G., Baltensberger U., and Prévôt A.S.H. specify that different forecast models match in the increase of emissions caused with maritime transport in the area of the European Atlantic coast and southern coast of Spain. On the other hand, the same models match in the decrease of emissions caused with maritime transport in the areas around the English Channel, North Sea, and Baltic Sea [8].

Other authors deal with the issue of transport proposal separately. For example, Chen, Pengfei, Huang, et al. introduce some options to utilise mathematical methods for planning a recommended route for autonomous vessels [9].

The impact of voyage velocity on particular transport proposals is dealt by Eide L., Ardal G.C.H., Evsikova N., Hvattum L.M., and Urrutia S. They give a closer look at the impact of the cargo volume and velocity on planning transport routes; however, their models do not significantly take into account the factor of the sustainability and impact on the environment [10].

Thus, the authors of this paper focus on the linking between these two important topics in the field of maritime transport. By now, the authors have not found any detailed studies and scientific publications that would link the evaluation of proposed transport routes with the environmental aspect of behaviour of order parties in maritime transport.

### 2.1. The Importance of Maritime Transport in Container Transport and Its Impact on the Environment

From the point of view of transporting cargo in containers, there currently exist many alternatives and solutions. Maritime transport represents the most advantageous alternative almost in all aspects when speaking about container transport between certain geographical units. This is especially true for transport between Europe and North America, Asia as well as Australia. Likewise, in other regions of the world, the influence of maritime transport on container transport increases progressively. This fact can mainly be justified with smaller costs of transport when compared to other kinds of transport. The next reason is the total volume of cargo, which can be carried by means of maritime transport

in comparison with means of other kinds of transport, mostly in transports for longer distances [11].

From the environmental perspective in case of intercontinental transports, maritime transport represents a highly effective way of container transport. When considering the total volume of cargo and the total transport distance, it still offers one of the greenest solutions of transport when recalculated per one intermodal transport unit. Thus, in this perspective, the importance of maritime transport clearly reflects both the development and the current state of container transport by sea and the estimate of the future development of container transport by sea [12].

Nowadays, the transport of containerised cargo happens in all regions of the world (Table 1). However, the major share in transport and transload of containers is mainly divided among countries in the territory of Asia, Europe, and North America (Figure 1). As stated in Table 2, the transload of containers is concentrated in the 20 biggest container ports worldwide. In 2018, more than 347.8 million TEU were transloaded through these terminals, which made up to 43.8% of the world's trade carried by sea. Ports in Asia, mostly Shanghai and Singapore, feature the leading position; in the long term, they hold the first two positions when speaking about container transloading. In Europe, only ports in Antwerp, Rotterdam, and Hamburg can be compared to those ports on the worldwide basis. In the territory of North America, the group of the biggest ports in the world include Los Angeles and Long Beach. Altogether, there are only five ports lying out of coastal Asia [12].

**Table 1.** The total volume of transported TEU in individual regions of the world and their percentage change in 2017–2018.

|  | **2017** | **2018** | **Percentage Change** |
|---|---|---|---|
| Africa | 30,398,569 | 30,940,898 | 1.8% |
| Asia | 488,852,650 | 510,513,120 | 4.4% |
| Europe | 119,359,397 | 125,888,633 | 5.5% |
| Latin America and the Caribbean | 48,863,196 | 51,669,025 | 5.7% |
| North America | 58,510,434 | 61,352,043 | 4.9% |
| Oceania | 12,003,344 | 12,896,887 | 7.4% |
| Total—The entire world | 757,987,590 | 793,260,606 | 4.7% |

Source: [12].

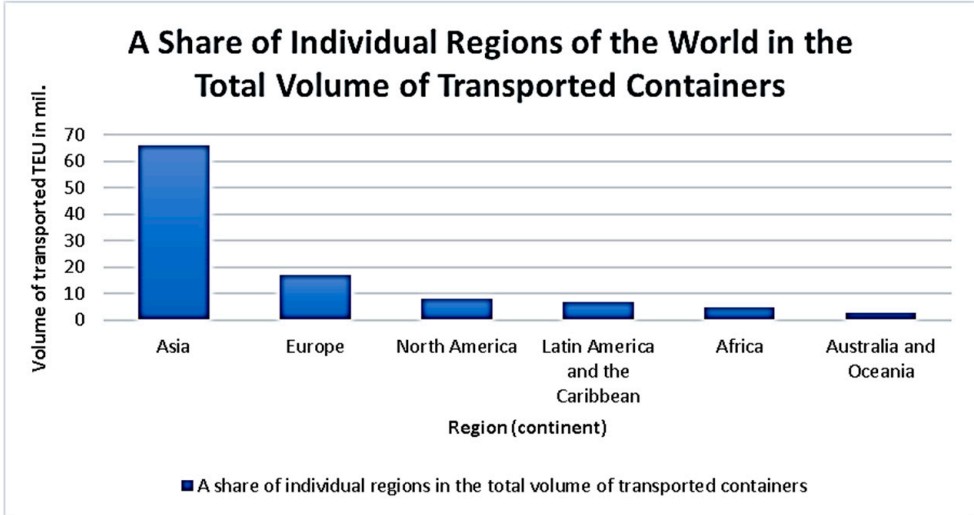

**Figure 1.** A share of individual regions of the world in the total volume of transported containers. Source: [12].

**Table 2.** The total volume of TEU transloaded in the 20 biggest container ports in the world.

|  | Throughput in 2018 | Annual Percentage Change in 2017–2018 |
|---|---|---|
| Shanghai | 42,010,000 | 4.4% |
| Singapore | 36,600,000 | 8.7% |
| Ningbo-Zhousan | 26,350,000 | 6.9% |
| Shenzen | 25,740,000 | 2.1% |
| Guangzhou | 21,920,000 | 7.6% |
| Busan | 21,660,000 | 5.5% |
| Hong Kong, China | 19,600,000 | −5.6% |
| Qingdao | 19,320,000 | 5.5% |
| Tianjin | 16,000,000 | 6.2% |
| Dubai | 14,950,000 | −2.9% |
| Rotterdam | 14,510,000 | 5.7% |
| Klang | 12,030,000 | 0.4% |
| Antwerp | 11,100,000 | 6.2% |
| Xiamen | 10,700,000 | 3.1% |
| Kaohsiung | 10,450,000 | 1.8% |
| Dalian | 9,770,000 | 0.6% |
| Los Angeles | 9,460,000 | 1.3% |
| Tanjung Pelepas | 8,790,000 | 6.4% |
| Hamburg | 8,780,000 | −0.2% |
| Long Beach | 8,070,000 | 3.7% |

Source: [12].

In each economic sector, there exist companies' efforts to minimise costs for the sake of a higher profit. Thus, in maritime transport, it has become a standard practice to register vessels in countries with low claims to enforce international regulations from the point of view of both ecology as well as fair working conditions for people from the third countries working on the vessels of these companies. It was the negligent attitude of some companies together with a neglect of safety and environmental standards that led to a series of serious sea accidents in the 1990s. Mainly, vessels sailing from areas around Africa and some parts of Asia and Latin America pose the risk. However, maritime transport undergoes some positive changes nowadays. The International Maritime Organisation, classification societies, and other organisations acting in maritime transport adopt a range of measures to ensure the sustainability and environment protection in maritime transport [13].

As to emissions, the usage of fuels with lower sulphur content is significantly promoted at international as well as national levels; ports in individual regions of the world tighten their rules for handling substances polluting seas and oceans [14].

Through a consistent adhering to these regulations and a constant control of their compliance in cooperation with a technical development in the field of a green operation of sea ships, it is possible to secure the sustainability of maritime transport even with a continually increasing volume of transported cargo [15].

### 2.2. The Estimate of a Possible Impact of the Future Development of Container Transport on the Environment

When presenting the advantages of maritime transport in container transport from the perspective of the sustainability and impact on the environment, there occurs one main issue—the way maritime transport deals with an increasing demand for container transport. The data from the period 1995–2018 (Figure 2) imply that an annual increase in the number of transported containers (with some exceptions) occurred there for the last approximately 25 years. Thus, from the point of view of evaluating a possible development of maritime transport's impact on the environment, first of all, it is required to know the most likely development of container transport by sea. For the sake of a better illustration and more detailed estimation with the impact on the European region, it is appropriate to select the closest transatlantic major transport route [16].

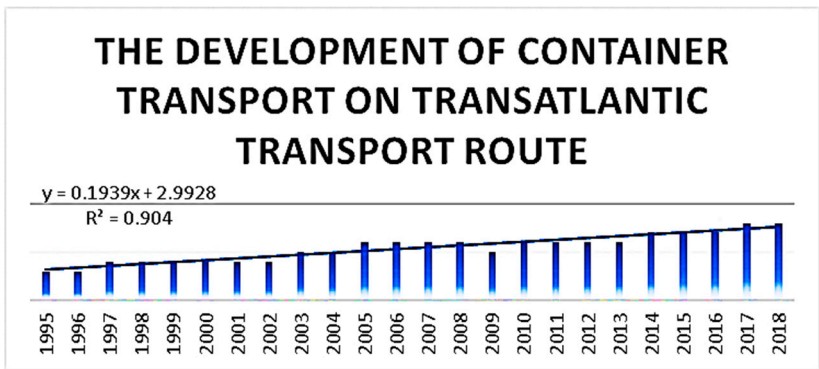

**Figure 2.** The development of container transport on transatlantic transport route in 1995–2018. Source: [1].

Since there is no need to know the exact values but rather a long-term trend of the development, it is appropriate to apply the method of least squares due to its simplicity and time undemandingness. The method can use a regression line to express a possible trend of the development of container transport on a given transport route. For this purpose, it is necessary to express an "approximation function" [16–18].

To determine the approximation [19] function, we come out from the measured values $((x_i, y_i), i = 0, 1, 2, \ldots, n)$, where the selected values of the measurement are represented with $x_i$ variable and the measured values $y_i$ represent the values of the function f(x) at point $x_i$. In that case, the resulting approximation function has the following form:

$$\varphi(x) = a_0 + a_1x \tag{1}$$

where

$\varphi(x)$—the value of the approximation function [piece, TEU...],
$a_0$, $a_1$ unknown constants [R].

The result of determining the estimation of the future development trend on the transatlantic transport route (Figure 3) is a curve that constitutes a graphical representation of the approximation function (Figure 2). This curve confirms an increase in container transport on the selected transport route, occurring there for the following four years. With an increasing number of transported containers, we can expect an increase in the loading capacity of container ships; this is also related to the volume of consumed fuels and released emissions. Thus, in the future, it can be anticipated that the trend development of container transport's impact on the environment will copy the trend of the container transport's growth on individual transport routes [19–21].

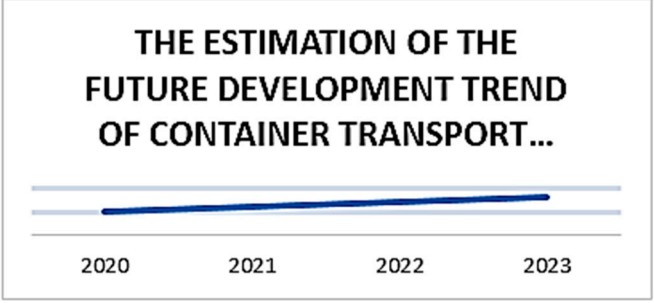

**Figure 3.** The estimation of the future development trend of container transport on the transatlantic transport route in 2020–2023. Source: Authors.

Likewise, many studies state that from the point of view of the total performance of vessels and installation of marine engines, the main share will mostly represent container ships (Table 3), and as noted in Table 3, the average operating hours of these vessels per

year is expected to reach 6700–7200 h/year [7].

**Table 3.** The summary of engine profiles from a manufacturer and operator survey.

| Ship Types | Installed Main Engines | Average Operating Hours per Year | Average Engine Load, % of Installed Power |
|---|---|---|---|
| Bulk carriers, tankers | two-stroke: 91%, four-stroke: 6% | 6500 | 55% if low freight rates, 80% if normal freight rates |
| Large container vessels (>1500 TEU) | two-stroke: 100% | 6700–7200 | 80% |
| Small container vessels (<1500 TEU) | two-stroke: 55%, four stroke: 45% | 6300–6700 | 80% |
| Crude oil carriers | two-stroke: 80%, four stroke: 19% | 6700–7300 | 75% |
| Lift-on Lift-off (LoLo) | two-stroke: 55%, four stroke: 32% | 6000 | 80% |
| Roll-on roll-off (Ro-Ro) | two-stroke: 11%, four stroke: 77% | 6500–7000 | 80–85% |
| Passenger vessels | primarily four-stroke | 4000 | 55% |
| Fishing vessels | two-stroke: 3%, four-stroke: 69% | 6700–7300 | 70% |

Source: [7].

Likewise, the published studies (Table 4) present that vessels for cargo transport will take a share in the production of carbonic oxides emissions to a lesser extent than vessels of other types. On the other hand, they will have the same share in the production of sulphur oxides and a slightly higher share in the production of nitrogen oxides [22–24].

**Table 4.** The fleet-average summary of in-service emissions factors (g/kWh).

| Ship Type | $NO_X$ | $SO_X$ | $CO_2$ [a] | HC | PM [b] |
|---|---|---|---|---|---|
| Cargo vessels | 17 | 10 | 655 | 0.6 | 1.25 |
| Non-cargo | 16 | 10 | 702 | 0.5 | 1.25 |
| Auxiliaries (kg/tonne) | 59 | 38 | 3180 | 1.8 | 2.50 |

[a] On the basis of an average fuel–sulphur content of 2.5% for heavy fuel oil, [b] PM factors for maritime diesels remain very uncertain, are still difficult to measure, and can be defined differently. Source: [7].

For the sake of the maritime transport sustainability, a constant technical and technological development in different areas of its operation is important. Without tightening the regulations and making new ones applicable to the environment in maritime transport, it is impossible to maintain a minimum impact of maritime transport on the environment from a long-term perspective and under a constantly increasing consumption of the population. From the point of view of the future development of maritime transport, there is a minimum requirement to maintain the current state and to sustain the fact that maritime transport is one of the greenest ways of transport when recalculated per 1 tonne of cargo transported for a certain distance [25–27].

### 2.3. The Importance of a Customer's View on the Application of Social Responsibility and Environment Protection Policies

The fundamental goal of many international companies that export their products by maritime transport is the minimisation of transport costs in order to achieve a higher profit. On the other side, in recent years as part of a propagation of companies, in business and the economy, the phrase "social responsibility" has been occurring more and more. One of the most significant aspects of this new business policy is the effort to ensure sustainability and a better environment not only in production but also in transport [6].

As part of this concept, companies adapt their operation to principles of the environment protection and to cooperation with local communities in order to improve the quality of life in locations and countries that their business is directly connected with [28–30].

As given in Table 5, many owners of container ships do currently apply principles of social responsibility and consideration for the environment. Mainly, it is a modernisation and reconstruction of vessels to reduce the ecological burden, a controlled demolition and recycling of retired vessels in countries that meet strict claims for consideration towards the environment and work force. Multiple researches also prove that the application of this business policy's principles makes them distinct from less responsible operators of container ships [6].

Table 5. The application of the sustainability and environment protection policy by operators of line maritime transport.

| CSR Concern | Total | Shipping Company | | | | | | | | | | | | | | |
|---|---|---|---|---|---|---|---|---|---|---|---|---|---|---|---|---|
| | | APM | MSC | CMA/CGM | NYK | MOL | K-LINE | OOCL | HMM | COSCO | H-LOYD | H-SUD | EVERGREEN | YMM | PIL | UASC |
| $CO_2$ emissions | 15 | • | • | • | • | • | • | • | • | • | • | • | • | • | • | • |
| Energy and fuel efficiency | 15 | • | • | • | • | • | • | • | • | • | • | • | • | • | • | • |
| Renewable energy | 5 | | | | • | • | • | • | | | | • | | | | |
| Ballast water | 15 | • | • | • | • | • | • | • | • | • | • | • | • | • | • | • |
| $SO_X$ emissions | 13 | • | | • | • | • | • | • | • | • | • | • | • | • | • | |
| $NO_x$ emissions | 11 | • | | • | • | • | • | • | • | • | | • | • | • | • | |
| Oil pollution | 13 | • | | • | • | • | • | • | • | • | • | • | • | | • | • |
| Ship demolition | 13 | • | • | • | • | • | • | • | • | • | • | • | • | | | • |
| General waste | 12 | • | | • | • | • | • | • | • | • | • | • | • | • | | |
| Anti-fouling paint | 11 | | • | • | | • | • | • | • | • | | | • | • | • | • |
| Noise control | 5 | | | | | | | • | • | • | • | • | | | | |

Source: [6].

## 3. Materials and Methods

The main objective of the paper is to analyse the impact of the sustainability and environment protection factor on the final choice of a shipping route that most suits the needs of the customer. Through the explanation of how this factor—representing the application of the sustainability and environment protection policy—can affect the final choice of a preferred route, this paper strives to point out new criteria of customers that can mean a competitive advantage of transport and shipping firms in the future.

It is necessary to follow several steps while analysing the impact of the sustainability and environment protection on the choice of the preferred shipping route. At the beginning, it is required to determine the fundamental method for the evaluation of proposals. In this case, it is suitable to apply one of the multi-criteria decision-making methods. The Analytic Hierarchy Process (AHP) method represents one of the most suitable methods because it allows evaluating proposals among each other as well as determining the weights of individual criteria [11]. Based on the objective evaluation of shipping and transport companies' customers, we will set basic decision factors, and each of them will be assigned respective weights to the highest objectivity level possible. Then, we will evaluate individual proposals without the environmental factor of transport and with this factor being applied. There are 2 possible results. The first one is that the overall ranking of proposals will not change, and the second one is that there will be a change in the recommendation of the transport route.

### 3.1. The Application of the Multi-Criteria Decision-Making Method to Evaluate Individual Proposals

Making decisions regarding advantages of particular alternatives for the sake of a selection often requires the most optimal alternative to fit multiple criteria at the same time. However, individual criteria may differ in their nature. The nature of these criteria can manifest qualitative as well as quantitative characteristics, e.g., price and quality. In case of evaluating the impact on the environment, one of these indicators that can significantly influence the final ranking of criteria is the ecological factor, which may, for instance, represent the type of fuel used in a vessel during its voyage. In many cases, a general task of the multi-criteria decision-making is classified by the way individual alternatives belonging to this set are specified. If this set contains one complex list of alternatives, then this evaluation can be referred to as a multi-criteria evaluation of individual variants [31,32].

However, if there are conditions that the most favourable alternative must meet, and these conditions create a set of permissible alternatives, then the task of multi-criteria programming (vector optimisation method) exists there. In this kind of problem, the individual alternatives are expressed as n-tuples of non-negative numbers, when all of them meet the set criteria and their number is not limited in any way. In case of criteria for the selection of the most appropriate proposal, their expression using objective functions is used. That implies that such an expression can be quantitative only [33,34].

### 3.2. Analytic Hierarchy Process (AHP) Method

One of the multi-criteria decision-making methods is the AHP (Analytic Hierarchy Process) method. The origin of this method is based on professor Saaty's work in 1980. The method arises from the knowledge that in order to solve a given decision problem, we need more than the information on all elements impacting the overall result of the analysis; the information on their mutual relations as well as intensity with which they effect each other in these relations is required, too. This perspective makes it possible to visualise this decision problem as a mutually ordered hierarchical structure (Figure 4). This structure comprises a certain number of levels, and individual levels contain several elements [11,19].

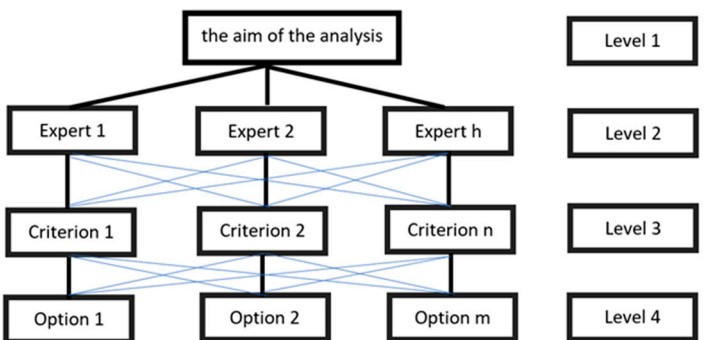

**Figure 4.** Multi-criteria decision-making levels using the Analytic Hierarchy Process (AHP) method. Source: [19].

However, if there are conditions that the most favourable alternative must meet, and these conditions create a set of permissible alternatives, then a task of a multi-criteria programming (vector optimisation method) exists there. In this kind of problem, the individual alternatives are expressed as n-tuples of non-negative numbers, when all of them meet the set criteria and their number is not limited in any way. In case of criteria for the selection of the most appropriate proposal, their expression using objective functions is used. That implies that such an expression can be quantitative only [11,19].

The first level represents the final aim of the analysis; the second level contains evaluations of experts who participate in the evaluation of individual criteria and proposals. Then, in the third level, individual criteria are evaluated, and in the last fourth level, individual variants are judged.

Individual levels are also characteristic in that in each of them, it is possible to determine relations among all its components. In case of a 4-level hierarchy, we will get one pairwise comparison matrix in total. In the third level, we will get matrices, the number of which depends on the evaluating experts, and in the fourth level, the resulting number will depend on the number of criteria. Using a mutual recalculation, weights of respective evaluating experts may be assigned to individual variants in these matrices. Thanks to this procedure, numerical values referred to as "preference indices for individual variants" can be obtained. In case we would like to express the evaluation of proposals (variants) from the perspective of experts as well as individual criteria, individual preference indices can be added from the point of view of all criteria being judged. To determine a preferred criterion and to determine the size of this preference for each pair of criteria, the "Method of Quantitative Pairwise Comparison", also referred to as Saaty's method by its developer, can be applied. To express the size of preferences in case of the Saaty's method application, a table with a point scale is used (Table 6). If the preferences are to be expressed with a greater sensitivity, even intergrades can be used (2, 4, 6, and 8) [19].

**Table 6.** Expression of size preferences by Saaty.

| Numerical Expression | Expression of Size Preferences |
|:---:|:---:|
| 1 | Criteria are of the same importance |
| 3 | The first criterion is slightly more important than the second one |
| 5 | The first criterion is strongly more important than the second one |
| 7 | The first criterion is very strongly more important than the second one |
| 9 | The first criterion is absolutely more important than the second one |

Source: [13].

If the AHP method is applied to decision-making processes, it is required to specify the consistency of the given matrix in order to correctly determine the comparison based

on criteria. The consistency index (CI) serves to judge the consistency. While judging the consistency, the CI index is divided with the random consistency index (referred to as RI) [11,19].

The RI index expresses the average of 500 matrices altogether, which were randomly generated during the pairwise comparison with a so-called Saaty's scale. The values of indices differ by the number of criteria being judged. In case there are 5 criteria judged, the average value of this index is 1.11 [11].

This way, the resulting consistency index will be obtained, which is referred to as CR:

$$CR = CI/RI.$$

If the resulting consistency ratio meets the condition CR > 1, that matrix can be considered consistent.

To judge the weights of individual criteria using the AHP method, multiple techniques can be used. The simplest and, at the same time, the most frequently used procedure is the expression using a geometric mean. In addition to this option, the calculation of individual weights of the matrix from the eigenvector (w) of the particular matrix can be applied as well. The eigenvector of the matrix can be derived from the following relation:

$$(S - \lambda\max I)w = 0 \tag{2}$$

where

$S$—Saaty's matrix
$\lambda_{max}$—The principal eigenvalue
$I$—a unit matrix
$W$—an eigenvector of the matrix.

This procedure will bring resulting weights using the following relation:

$$v_j = \frac{w_j}{\sum_{j=1}^{n} w_j}, \ j = 1, 2, \ldots, n. \tag{3}$$

Regarding the complexity and time demands of this procedure to set individual weights, multiple simplified procedures are often used, e.g., totals in individual rows of an S matrix, an expression of a reciprocal of totals of individual matrix columns, a division of each column with the total of that column, or using the *k*-th root of a product of elements in individual rows of an S matrix [11,19].

It is recommended to sort the weights and values of individual criteria into tables. It is appropriate to assign the weights to individual criteria based on customer's preferences or actual preferences of the company, realising the transport [11,19].

*3.3. A Dominance Rule*

In some cases, a more appropriate procedure can be set based on common consideration. Most of all these are cases when it is evident that one of the introduced alternatives univocally dominates the other ones. For such a type of judgment, a so-called "dominance rule" can be applied.

The dominance rule belongs to basic rules while using multi-criteria decision-making methods. Its purpose is to roughly judge individual alternatives while solving the given problem. As part of this rule, we judge which alternative dominates another alternative, and which alternative is "dominated" and which one is "non-dominated" [19].

A dominated alternative represents a solution to which a better alternative within the existing alternatives portfolio may be found. However, in this case, the fundamental condition is as follows: this alternative must be worse than the other ones by at least one criterion, and at the same time, it cannot be better than the alternative referred to as non-dominated by any criteria [19].

A non-dominated alternative represents the best possible solution ever to which there exists no better alternative. When compared to a dominated alternative, the non-dominated alternative is better by at least 1 criterion, and it is not worse than the dominated one by any of selected criteria (Managerial decision making) [19].

## 4. Results

The methodology described above has been applied to a case study that deals with the transport of a standardised container, series ISO 1A, between Europe and North America. Proposals within the case study have been judged using the selected multi-criteria decision-making method. The proposals come from materials of the shipping company, and individual criteria match the most frequent requirements of parties ordering the procurement of transport by the shipping company. The evaluation and weights of individual criteria are based on the survey conducted among 10 selected customers of the shipping company. The case study should result in a comparison of the final proposal for the customer without and with the application of the criterion, i.e., the environmental factor of transport. Following this result, it is possible to express how that criterion can affect the setting of the recommended proposal according to specific requirements of the customer.

### 4.1. The Utilisation of the AHP Method in Setting the Recommended Route without and with the Application of the Factor of the Impact on the Environment—A Case Study

Based on the materials of the shipping company, there are two proposals for the specific container transport between Nitra and Los Angeles port available. Proposal No. 1 (Figure 5) assumes that the container will be transported by sea via Hamburg port to Los Angeles port through the transatlantic transport route and the Panama Canal. Proposal No. 2 (Figure 6) comes out from the assumption the transport will lead via Koper port and Suez Canal and then through the Shekou and Geelong ports to Los Angeles.

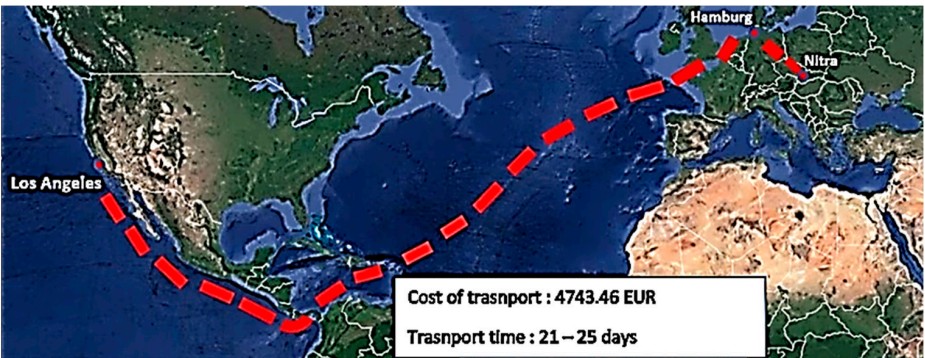

**Figure 5.** The proposal for transport of a standardised container, series ISO 1A, by the Transatlantic transport route. Source: Google Maps.

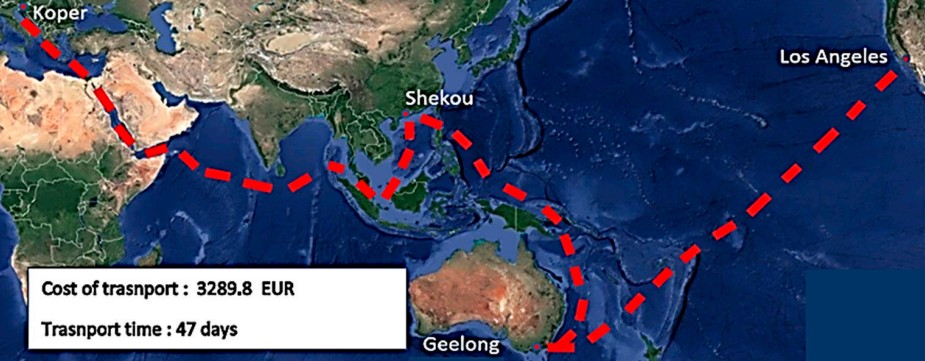

**Figure 6.** The proposal for transport of a standardised container, series ISO 1A, by the main shipping route of Europe–Asia and by the Transatlantic transport route. Source: Google Maps.

The final price for transport, total transport time, number of ports during the voyage, number of passes through a canal, and number of estimated risks during transport were taken for the main criteria in this case. The result is the mutual comparison of individual routes as well as the comparison of the impact of transport ecological aspect on the final setting of the recommended route per the customer's preferences.

For the mutual comparisons of routes as well as for the comparison of the impact of the transport ecological aspect, the same multi-criteria decision-making method was applied.

In case of proposal No. 1, the container is transported on the deck of the vessel operated by the company that applies the policy of sustainability, social responsibility, and environment protection. The company applies stringent regulations regarding the disposal and recycling of retired vessels, it uses fuels with a low content of sulphur, and it applies a stringent policy for handling wastes that arise during the vessel operation.

In proposal No. 2, the acting company operates a vessel under the flag of the country that is located in the territory of Africa to minimise costs of fees and to keep minimum international standards in the area of the environment. The company usually looks for opportunities to apply lower standards for handling with waste products in order to reduce costs. However, the vessel of this company meets all international standards required by the International Maritime Organisation (IMO) and the MARPOL regulation.

*4.2. The Setting of a Recommended Proposal for a Customer without Consideration of the Factor of the Impact on the Environment*

The results of the survey of preferences of the addressed shipping company's customers point out a certain ranking of factors on the basis of which the customers make decisions regarding whether they will accept a given offer or not. Altogether, there were 10 customers addressed; they evaluated criteria such as the price for transport, total transport time, number of estimated risks during the voyage, necessity to pass through a canal, or number of stops in ports during the voyage. The addressed customers compared individual criteria among each other, and they used the scale 1–9, in which 1—equally important, 9—9 times more important than the comparing factor (Saaty's method—table of preference sizes). The most frequently occurring comparisons created the final comparison of individual criteria of the customer of the addressed shipping company, based on which the weights of individual criteria were set (Table 7). From the perspective of a customer who does not apply the policy of sustainability and environment protection, the most important criterion is the price for transport. Therefore, it has the highest importance rating in the face of other parameters. In case of weights (importance) assignments, it means that the price for transport is three times more important than the total transport time, five times more important than estimated risks, seven times more important than passage through a canal, and nine times more important than the number of the vessel stops in ports during the voyage.

The second rating belongs to the total transport time, which is evaluated by this customer as a more important criterion than the number of passages through a canal (nine times), estimated risks (seven times), and the number of stops in individual ports during the voyage (five times). The price and total transport times feature the highest ratings mainly due to a possible reduction of costs in order to achieve a higher profit, and fast delivery of goods to achieve a fast financial return.

Risks represent some possible complications and related losses. That is why the customer considers the judgement of possible risks as more important (three times) than the number of stops in ports during the voyage or passages through a canal.

There are two factors in the last ranking; under favourable conditions, they do not significantly impact the price or the total transport time. If all deadlines for loading and unloading in individual ports are kept and the sailing situation in canals does not create conditions for an additional delay, these two factors may be considered less important in the evaluation from the customer's point of view.

**Table 7.** Evaluation and weights of individual criteria (by Saaty).

| | Price for Transport | Total Transport Time | Estimated Risks | Passage through a Canal | Number of Stops in Ports during the Voyage | Weight |
|---|---|---|---|---|---|---|
| **Price for transport** | 1 | 3 | 5 | 7 | 9 | 0.4937 |
| **Total transport time** | 1/3 | 1 | 7 | 9 | 5 | 0.3182 |
| **Estimated risks** | 1/5 | 1/7 | 1 | 3 | 3 | 0.0956 |
| **Passage through a canal** | 1/7 | 1/9 | 1/3 | 1 | 3 | 0.0548 |
| **Number of stops in ports during the voyage** | 1/9 | 1/5 | 1/3 | 1/3 | 1 | 0.0378 |

Source: authors.

In case of comparison of individual proposals and based on individual criteria, their weights and evaluations were set equally on the basis of a survey. Again, 10 customers evaluated two proposals using Saaty's table of preference sizes expressions; they were informed of the price for transport, the total transport time, the number and naming of risks being estimated in case of transport in the given route, whether the transport would lead through a ship canal, and the total number of ports where the vessel was loaded and unloaded. The most frequent evaluations were determined from the perspective of each factor; they represent the final comparison of proposals from the point of view of individual factors.

When looking at the price for transport (Table 8), most of the addressed customers indicated that proposal No. 2 was seven times more significant than proposal No. 1.

**Table 8.** Evaluation and weights of individual proposals (by Saaty) based on the price for transport.

| Price for Transport | Proposal No. 1 | Proposal No. 1 | Weight |
|---|---|---|---|
| Proposal No. 1 | 1 | 1/7 | 0.125 |
| Proposal No. 2 | 7 | 1 | 0.875 |

Source: authors.

When looking at the total transport time (Table 9), most of the addressed customers indicated that proposal No. 1 was nine times more significant than proposal No. 2.

**Table 9.** Evaluation and weights of individual proposals (by Saaty) based on the total transport time.

| Total Transport Time | Proposal No. 1 | Proposal No. 1 | Weight |
|---|---|---|---|
| Proposal No. 1 | 1 | 9 | 0.9 |
| Proposal No. 2 | 1/9 | 1 | 0.1 |

Source: authors.

When looking at the number and naming of risks being estimated (Table 10), most of the addressed customers indicated that proposal No. 1 was five times more significant than proposal No. 2.

**Table 10.** Evaluation and weights of individual proposals (by Saaty) based on the estimated risks.

| Estimated Risks | Proposal No. 1 | Proposal No. 1 | Weight |
|---|---|---|---|
| Proposal No. 1 | 1 | 5 | 0.8333 |
| Proposal No. 2 | 1/5 | 1 | 0.1667 |

Source: authors.

When looking at the necessity to pass through a ship canal (Table 11), all addressed customers indicated that proposal No. 1 was equally significant as proposal No. 2.

**Table 11.** Evaluation and weights of individual proposals (by Saaty) based on the passage through a ship canal.

| Passage Through a Canal | Proposal No. 1 | Proposal No. 1 | Weight |
|---|---|---|---|
| Proposal No. 1 | 1 | 1 | 0.5 |
| Proposal No. 2 | 1 | 1 | 0.5 |

Source: authors.

When looking at the number of stops in ports during the voyage (Table 12), most of the addressed customers indicated that proposal No. 1 was five times more significant than proposal No. 2.

**Table 12.** Evaluation and weights of individual proposals (by Saaty) based on the number of stops in ports during the voyage.

| Number of Stops in Ports during the Voyage | Proposal No. 1 | Proposal No. 1 | Weight |
|---|---|---|---|
| Proposal No. 1 | 1 | 5 | 0.8333 |
| Proposal No. 2 | 1/5 | 1 | 0.1667 |

Source: authors.

Weights from the mutual comparison of criteria among each other and weights set on the basis of comparison of individual proposals by particular criteria determine a sum of evaluations that can be obtained as a sum of products of an individual proposal's weight with the respective criterion's weight (Table 13). For example, in case of proposal No. 1, it is as follows: (0.125 * 0.4937) + (0.9 * 0.3182) + (0.8333 * 0.0956) + (0.5 * 0.0548) + (0.8333 * 0.0378) = 0.4866. Comparing the sum of evaluations, we will get the final ranking of proposals (a bigger value—a better result from the perspective of ranking).

**Table 13.** The final determination of proposals ranking by individual criteria.

| | Price for Transport | Total Transport Time | Estimated Risks | Passage through a Canal | Number of Stops in Ports during the Voyage | Evaluations Sum | Overall Ranking |
|---|---|---|---|---|---|---|---|
| Proposal No. 1 | 0.125 | 0.9 | 0.8333 | 0.5 | 0.8333 | 0.4866 | 2 |
| Proposal No. 2 | 0.875 | 0.1 | 0.1667 | 0.5 | 0.1667 | 0.5071 | 1 |
| Criteria weights | 0.4937 | 0.3182 | 0.0956 | 0.0548 | 0.0378 | | |

Source: authors.

On the basis of a mutual comparison of individual proposals without consideration of the ecological aspects of transport (Table 13), it can be concluded that the proposal for transport via Koper port to Los Angeles port is more favourable for a customer who mostly prioritises the price. However, no proposal can be considered dominant, because in a closer view, we will find out that the price, as a decisive factor, does not significantly influence the choice of a recommended route for the customer, although it impacts the overall ranking. In case of the recommended route (proposal No. 2) and alternative route (proposal No. 1), the total difference in the sum of point evaluations is approximately 0.02 only. This result is mainly influenced with the second significant factor—the total transport time. This parameter together with other parameters (risks, canals, and the number of stops in ports) settles the point evaluation in a certain way. One of the reasons is also the dominance rule, which could be applied if the price was not the criterion with the biggest weight in this case. Thus, even though a certain ranking is set, we cannot speak about the definiteness of a single proposal.

*4.3. The Setting of a Recommended Proposal for a Customer with Consideration of the Factor of the Impact on the Environment*

In this case, the evaluations of individual criteria by Saaty's method (Table 14), based on which their weights were set, represent the results of the survey from the previous comparison. Likewise, the comparison of individual proposals among each other by individual criteria matches the comparison in the previous case. The only difference is the addition of one factor, namely the ecological aspect of transport. The last point of the survey conducted at the addressed shipping company among its customers was the evaluation of importance provided the customer would constantly adhere to the policy of sustainability, social responsibility, and environment protection. In this light, most of the customers would evaluate the ecological aspect of transport as an absolutely more significant factor than the price for transport (evaluation No. 9), the total transport time and the number of stops in ports as a very strongly more significant factor (evaluation No. 7), and the estimated risks and the number of passages through a ship canal as a strongly more significant factor (evaluation No. 5).

**Table 14.** Evaluation and weights of individual criteria (by Saaty).

| | Ecological Aspect of Transport | Price for Transport | Total Transport Time | Estimated Risks | Passage through a Canal | Number of Stops in Ports during the Voyage | Weight |
|---|---|---|---|---|---|---|---|
| **Ecological aspect of transport** | 1 | 9 | 7 | 5 | 5 | 7 | 0.4854 |
| **Price for transport** | 1/9 | 1 | 3 | 5 | 7 | 9 | 0.2235 |
| **Total transport time** | 1/7 | 1/3 | 1 | 7 | 9 | 5 | 0.1616 |
| **Estimated risks** | 1/5 | 1/5 | 1/7 | 1 | 3 | 3 | 0.0627 |
| **Passage through a canal** | 1/5 | 1/7 | 1/9 | 1/3 | 1 | 3 | 0.0394 |
| **Number of stops in ports during the voyage** | 1/7 | 1/9 | 1/5 | 1/3 | 1/3 | 1 | 0.0274 |

Source: authors.

Then, in the second case, it is obvious that the customer actively applies all alternatives of the policy of sustainability, social responsibility, and environment protection. From the point of view of this customer, the ecological aspect of cargo transport represents the most important factor. The comparison of other factors among each other and the determination of their weights come out from the previous comparison based on the survey of the shipping company.

The term "ecological aspect of transport" may mean different impacts of cargo transport by sea on the environment, for example, the volume of sulphur oxides and carbon oxides emissions, the handling with ballast tanks fill, waste management, the ship-owners policy regarding the disposal of retired vessels, etc. Of course, when speaking about the customer's management, the order of importance of other rating factors remains basically the same.

In its last part, the survey evaluated two proposals by the ecological aspect of transport from the customer's perception. The data on individual proposals were added with information on the application of policies of social responsibility and environment protection by the operator of the container ship that transports the cargo (Table 5). The name of the company was not mentioned; only the information of whether the operator utilised all 11 items or not was provided. In case of the first proposal, the operator (ship-owner) actively applies all 11 items from Table 5. In case of the second proposal, the provider does not apply all 11 items from Table 5. Based on these parameters, the addressed customers indicated proposal No. 1 as five times more significant than proposal No. 2 in the survey from the point of view of their final decision (Table 15).

**Table 15.** Evaluation and weights of individual proposals (by Saaty) based on the ecological aspect of transport.

| Ecological Aspect of Transport | Proposal No. 1 | Proposal No. 1 | Weight |
|---|---|---|---|
| Proposal No. 1 | 1 | 5 | 0.8333 |
| Proposal No. 2 | 1/5 | 1 | 0.1667 |

Source: authors.

The remaining evaluations of proposals among each other are the same as in the first comparison without consideration of the factor of ecological aspect of transport, since the sample of customers is the same.

Weights from the mutual comparison of criteria among each other and weights set on the basis of comparison of individual proposals by particular criteria again as in case of the first comparison determine a sum of evaluations that can be obtained as a sum of products of individual proposal's weight with the respective criterion's weight (Table 16):

**Table 16.** The final determination of proposals ranking by individual criteria.

| | Ecological Aspect of Transport | Price for Transport | Total Transport Time | Estimated Risks | Passage through a Canal | Number of Stops in Ports during the Voyage | Evaluations Sum | Overall Ranking |
|---|---|---|---|---|---|---|---|---|
| Proposal No. 1 | 0.8333 | 0.125 | 0.9 | 0.8333 | 0.5 | 0.8333 | | 1 |
| Proposal No. 2 | 0.1667 | 0.875 | 0.1 | 0.1667 | 0.5 | 0.1667 | | 2 |
| Criteria weights | 0.4854 | 0.2235 | 0.1616 | 0.0627 | 0.0394 | 0.0274 | | |

Source: authors.

(0.8333 * 0.4854) + (0.125 * 0.2235) + (0.9 * 0.1616) + (0.8333 * 0.0394) + (0.5 * 0.0394) + (0.8333 * 0.0274) in case of proposal No. 1. Comparing the sum of evaluations, we will get the final ranking of proposals (a bigger value—a better result from the perspective of ranking).

On the basis of a mutual comparison of individual proposals with consideration of the factor of the ecological aspect of transport (Table 16), it can be concluded that the proposal for transport via Hamburg port to Los Angeles port is more favourable for a customer who mostly prioritises the ecological aspect of transport. In this case, the proposal can be considered dominant because in a closer view, we will find out that the ecological aspect of transport, as a decisive factor, significantly impacts the overall ranking and significantly influences the choice of a recommended route for the customer. In case of the recommended route (proposal No. 1) and the alternative route (proposal No. 2), the total difference in the sum of point evaluations is more than 0.6, whereas in the first comparison, i.e., without consideration of the ecological aspect of transport, it was approximately 0.2 only. It means that the application of the factor of the ecological aspect of transport may considerably deepen the difference between individual proposals. Likewise, upon this customer's requirement application, the resulting recommendation by the transport or shipping company will change. It proves that along with the arrival of new company policies and their active application to different stages of production and distribution, the transport and shipping companies must realise that they can gain a competitive edge through an appropriate judgement in the future, mostly in long-term business partnerships.

## 5. Discussion

The multi-criteria decision-making method can be used to compare individual proposals for transport from different aspects. The AHP method can be applied to compare not only individual proposals but also individual criteria among each other. When a certain criterion is added, the overall ranking of individual proposals can be changed. Weights of individual criteria play a critical role to a large extent. If one proposal dominates in all criteria except for one, it can lead to an adjustment of the overall ranking while determining the maximum weight of a certain criterion according to which that proposal is less favourable.

Alternatively, a sufficient weight of one criterion, even though it does not belong to the dominating ones, can change the overall ranking of alternatives and thus influence the final proposal for the customer. This phenomenon can be observed in both comparisons. In the first comparison, when the ecological aspect of transport was not considered, the price—as the most important aspect—managed to balance remaining criteria [35]. If the price for transport was not the decisive criterion, the dominance rule would univocally be applied. In the second comparison, we can see the addition of a new criterion, namely the ecological aspect of transport. If the most significant criterion from the first comparison—price—is assigned a smaller weight, and if, at the same time, a parameter with a weight higher than that of price is added—in the specific case, the ecological aspect of transport, it will change the overall ranking and cause a higher disproportion in the resulting evaluation of individual proposals. This knowledge is important mostly for the determination of the proposal by the individual needs of the customer. Currently, there are efforts in many companies to apply procedures and choose alternatives in line with a sustainable development and in consideration of the environment. This new corporate policy represents a new way of thinking of the company, which is effectively included into the marketing environment. It is the utilisation of transport with a minimum impact on the environment that will play a key role for these companies while choosing an offer presented by shipping and transport companies. Thus, the factors of ecology and sustainability will appear in the judgement of individual alternatives not only by customers but also by experts in the issue [35]. In order to conduct an independent assessment of the individual criteria, 10 entities were contacted that assessed the individual criteria. The entities were selected according to the geographical area of their business activities, according to the direction of exports and the orientation towards the application of the policy of sustainability and environmental protection. According to these criteria, entities operating on the international market were selected. The priority was focussed on multinational companies with a developed strategy of sustainability and environmental protection policy. The development of this plan by a company was a basic indicator that influenced the final choice of the preferred transport route. Therefore, the elaboration of the strategy for the development of sustainability and socially responsible business was a key factor of entity selection. The number of addressed entities (10 customers) represents a relevant sample of companies that actively fulfil the criteria of international operation and are also willing to participate in sharing information about application of their internal company policy. For more detailed research, it would be acceptable to apply a wider sample of entities. On the other hand, the currently applied principles of companies are mostly focussed on profit. So, authors have to admit that creating a larger sample of entities pursuing a policy of sustainability and environmental protection operating in the international sphere would not be successful. We all believe that more and more companies in international trade will not focus their priorities on reducing costs and increasing profits, but also on social responsibility and environmental protection. These companies already exist, but the number of these companies is still lower than the number of entities that take this factor into account. Therefore, this fact significantly affects the number of subjects that have been contacted. Ultimately, the inclusion of this factor—as proved with the case study in this paper—can have an impact on the customers' preferences as well as on the way of selecting, proposing and directing transport flows in connection with an increasing volume of cargo being transported, as it is estimated on the basis of the method of least squares. Thus, in general, this case study fastens the attention on the fact that if the transport and shipping company wants to be competitive, it must actively react to new corporate policies of sustainability and environment protection, too.

A new aspect of container transportation will probably be the epidemiologic situation in the world. From this point of view, the drop of volume of cargo being transported by sea is not at risk due to health hazards but rather economic hazards caused with a health crisis. However, recently, the stabilisation of economies and investment planning could have been observed. It means that the current economic state caused by the COVID-19 disease spread will be stabilised in the near future. Still, it is important to realise the

impact of the following epidemics on economic consequences in the production and distribution. From this perspective, it can again be observed that maritime transport will play a key role thanks to its offer of a low price for transport and separateness in cargo transport. There exist unquestionable advantages in case of container transport, even from the perspective of health and epidemic safety. Intermodal transport units are closed during the entire transport, and their handling is secured exclusively by means of machinery virtually in all stages. Thus, any transmission during transport is excluded; from the point of view of transport time for big distances (often more than 2 weeks), there exists a minimum risk also from the point of view of viruses' existence inside of intermodal units. The influence of the pandemic on the sustainability management of international shipment flow probably will not be so radical, because this issue has caused serious problems mainly in the transportation of passengers. If the economic situation is better, there will be a lot of opportunities regarding how to improve processes and how to improve our transportation system. So, we can hope that the environmental factor will be taken into account. This is probably the most significant impact of COVID on environmental protection in maritime transport.

The authors strongly believe that the results of this research confirm the importance of new strategies and the implementation of these strategies to the final recommendation for customers of freight forwarders. The environmental factor could cause the final proposal to be dramatically different when we accept new criteria (for example, environmental criteria). The positive effect of these results is that it brings new insight into this issue. The authors strongly believe that the environmental aspects and sustainability development concern not only shipping companies: it starts in route planning.

## 6. Conclusions

The increasing volume of cargo transported in containers causes a considerable pressure on the effectivity, sustainability, and ecology of transport. The overall trend of the future development univocally suggests that the container transport by sea will continue to grow. Many companies and organisations have actively adopted the policy of social responsibility and sustainability. These companies include more than operators of sea vessels, they also include their customers and customers of shipping companies. Using the methods of multi-criteria evaluation of transport alternatives, the occurrence of offers that are not suitable from the perspective of preferences in connection with the environment protection can be optimised. The inclusion of the impact on the environment into the evaluation will result in the elimination of non-ecological transports by sea as well as in the adaptation of the maritime transport market to new ecological thinking of transport order parties.

To study the impact of sustainability and environmental thinking of customers, it is necessary to proceed in several fundamental steps. First of all, the basic method for assessing individual alternatives must be determined; here, the possibility to assess individual factors among each other must be taken into account. Then, it is advantageous to assess individual alternatives among each other firstly without considering the ecological aspect of transport, and afterwards with considering the aspect. After all, the paper has outlined a change that may happen and pointed out its significance from the perspective of alternatives offered.

The comparison of results of the overall ranking of proposals without and with consideration of the ecological aspect of transport represents the aim of this paper. In other words, the paper looks at how adding the environmental aspect may change the resulting determination of the recommended route per the customer's wishes. It points out the importance of individual requirements of the customer in connection with the application of the sustainability and environment protection policy while determining a suitable offer of a transport route. This way, the paper attempts to clarify how these aspects can impact the customer's decision to accept or reject the given offer.

As the case study indicates, the addition of one distinct factor can change the overall ranking of importance of presented proposals. An equally essential finding is that a significant impact of this factor can also change the difference in the resulting sum of evaluations of individual proposals and radically remove the dominance of one of proposals. If the impact of transport sustainability and application of policies of social responsibility and environment protection represent this aspect, it can mean an essential change in transport routes planning. In the future, along with increasing claims for the environment, the impact of this factor will grow, and the adaptation to these changes will play one of the key roles in the area of transport and shipping.

However, when speaking about new topics regarding the impact on transport planning we cannot forget other equally significant factors that are still not taken into account enough in many companies nowadays, but their importance will grow in the future. Mostly, these factors are economic–political ones caused with trade wars, which have considerably been applied mainly by the USA and China in recent years, but the European Union and countries of Near East and Middle East have joined them to a certain extent, too.

COVID-19, which started to spread in the world at the beginning of 2020, also hit the sector of maritime transport. In the history of human beings, it is not the first time when maritime transport has been hit by this type or a similar disease. Scientists believe that the situation will change soon in the world after the worldwide vaccination process that has already been started will be completed.

The relevance of these factors, similar to the relevance of ecology and sustainability, will be a key value in the future; thus, it is appropriate for transport and shipping companies to start noticing these changes as soon as possible so that they can keep their importance in leading rankings in transport market in the future. After 2000, the International Maritime Organisation issued some regulations focussed on the reduction of the gases that cause the greenhouse effect. The next research will be focussed on the reduction of these gases, using alternative fuels and new types of engines.

The results of final comparison could contribute to the sustainability of maritime transport. Thanks to this comparison, it is possible that new freight forwarders implement this factor into their internal systems. This new type of thinking will create a pressure on shipping companies to concern new technologies of sustainability and environmental protection. This could be the main contribution of this paper and this research to the sustainability development of services in maritime transport. The authors also strongly believe that this comparison could have an effect on the flow of containers, because the implementation of environmental policy could change the current system of main shipping routes. These changes will be a challenge for local governments, international institutions, and shipping companies and also for the largest container seaports in the world.

**Author Contributions:** P.M. and P.B. conceived and designed the calculations; P.M. and J.L. analysed the data; A.D. and P.M. wrote the paper. All authors have read and agreed to the published version of the manuscript.

**Funding:** The paper is supported by the VEGA Agency by the Project 1/0798/21 "The Research on the Measures to Introduce Carbon Neutrality in the Rail and Water Transport" that is solved at Faculty of Operation and Economics of Transport and Communications, University of Zilina.

**Institutional Review Board Statement:** Not applicable.

**Informed Consent Statement:** Not applicable.

**Data Availability Statement:** Not applicable.

**Conflicts of Interest:** The authors declare no conflict of interest.

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
