# Peer review of "The Impact of an Environmental Way of Customer’s Thinking on a Range of Choice from Transport Routes in Maritime Transport"

_sustainability, doi:10.3390/su13031230_

Round 1

Reviewer 1 Report

Please check the equation some indexes are wrong!

Please check the literature - too many SUSTAINABILITY references, this could cause high selfcitation for the journal

Please consider implementing these papers:

Zalacko, R., Zöldy, M., & Simongáti, G. (2020). COMPARATIVE STUDY OF TWO SIMPLE MARINE ENGINE BSFC ESTIMATION METHODS. Brodogradnja: Teorija i praksa brodogradnje i pomorske tehnike71(3), 13-25.

Ficzere, P., Ultmann, Z., & Török, Á. (2014). Time–space analysis of transport system using different mapping methods. Transport29(3), 278-284.

Szabó, Z., Török, Ár., & Sipos, T. (2019). Order of the cities: Usage as a transportation economic parameter. Periodica Polytechnica Transportation Engineering.

Author Response

Reference to Review 1:

We would like to thank the reviewer for his insight into the subject addressed.

Please check the equation some indexes are wrong!

We have checked indexes and have improved the mistakes (especially in max ).

Please check the literature - too many SUSTAINABILITY references, this could cause high self-citation for the journal

We have deleted some references from SUSTAINABILITY to prevent high self-citation of this journal.

Please consider implementing these papers:

Zalacko, R., Zöldy, M., & Simongáti, G. (2020). COMPARATIVE STUDY OF TWO SIMPLE MARINE ENGINE BSFC ESTIMATION METHODS. Brodogradnja: Teorija i praksa brodogradnje i pomorske tehnike71(3), 13-25.

Ficzere, P., Ultmann, Z., & Török, Á. (2014). Time–space analysis of transport system using different mapping methods. Transport29(3), 278-284.

Szabó, Z., Török, Ár., & Sipos, T. (2019). Order of the cities: Usage as a transportation economic parameter. Periodica Polytechnica Transportation Engineering.

Three references from SUSTAINABILITY journal were replaced by these sources (of course, according to the content). 

Reviewer 2 Report

  1. The abstract provides general information, more information about the article material is missing.
  2. The introduction also provides general information, lacks more information about the essence of the article, emphasizes the analysis of the problem (topic).
  3. Please check the citation requirements. The names of the authors and references to the sources are also mentioned in some places - is this not a rule of double citation?
  4. In my opinion, subsections 3.1, 3.2, 3.3, 3.4, and 3.5 are a review of the literature, and should therefore logically be combined with Chapter 2 of the Literature review. Also, to the extent possible, in the above-mentioned subsections, update the data and use the 2019.
  5. Figure 3 needs to be updated because you write in it that it is “... Future Development Trend ....” and you are giving forecasts from 2019, when these data should already be clear and the forecast should start at least from 2020.
  6. In my view, there is an error in the explanation of the elements of formula 2, i.e. At line 329, technical element (compared to formula) naming error.
  7. It is not clear on what basis the 10 customers were selected. What could be taken and how were they selected? Is this a sufficient number of respondents, etc.? This type of information should be expanded either in the methodology section or to the results of the presented study.
  8. I missed the larger contribution of the authors in the analysis of results. That is, the opinion of the authors would be needed, for a deeper analysis, for example, what could have determined the results obtained, how they could influence the issue under consideration or other factors under consideration, and so on.
  9. It is recommended to adjust the discussion section, as there is more focused on the statement of facts from literature sources than what should be according to the requirements of the journal.
  10. The conclusions section also did not respond to the journal's requirements, and it also contains new facts that were not addressed in the article itself (eg COVID-19). There is also a lack of clear guidelines (directions) for possible further research.

Author Response

Reference to Review 2:

We would like to thank the reviewer for his insight into the subject addressed.

The abstract provides general information, more information about the article material is missing. 

- The abstract has been completely revised – as required by the reviewers. Now, it contains more information about the materials and methods used in this paper.   

The introduction also provides general information, lacks more information about the essence of the article, emphasizes the analysis of the problem (topic).    

- New information about environmental policy, analytic methods and the impact of environmental criteria on the final choice customers has been added. According to this information we are trying to emphasise the importance of customer's thinking for freight forwarders that emphasizes the main topic of this paper.           

Please check the citation requirements. The name of authors references to the sources are also mentioned in some places – is this not a rule of double citation? 

- We have made a revision and have deleted the authors mentioned in literature review from the list of sources. We have not found other examples of rule of double citation

In my opinion, subsections 3.1, 3.2, 3.3, 3.4, and 3.5 are a review of the literature, and should therefore logically be combined with Chapter 2 of the Literature review. Also, to the extent possible, in the above-mentioned subsections, update the data and use the 2019.

- We have completely moved the chapters 3.1, 3.2, 3.3, 3.4, and 3.5 from the part Materials and methods to the literature review. These chapters are separated from literature review by using new subchapters but these subchapters are logically connected to the literature review. On the other hand we are afraid that the part „Materials and methods“ is not adequate or too short and this change can affect the meaning of this part of the paper.     

Figure 3 needs to be updated because you write in it that it is “... Future Development Trend ....” and you are giving forecasts from 2019, when these data should already be clear and the forecast should start at least from 2020

- Figure 3 has been remade, as suggested by the reviewer 2.

In my view, there is an error in the explanation of the elements of formula 2, i.e. At line 329, technical element (compared to formula) naming error.

- We have to accept that it was a naming error. Better name of element was added.

It is not clear on what basis the 10 customers were selected. What could be taken and how were they selected? Is this a sufficient number of respondents, etc.? This type of information should be expanded either in the methodology section or to the results of the presented study.

- We believe that it could be better to add this part of the paper into the part called discussion. In this part there is also another information that is interrelated with these 10 customers. We have been trying to explain the number of addressed customers and also explain the main idea of this research related to addressed customers. 

I missed the larger contribution of the authors in the analysis of results. That is, the opinion of the authors would be needed, for a deeper analysis, for example, what could have determined the results obtained, how they could influence the issue under consideration or other factors under consideration, and so on.

The contribution of the authors has been added into discussion and conclusion.

It is recommended to adjust the discussion section, as there is more focused on the statement of facts from literature sources than what should be according to the requirements of the journal.

This part has been changed by the authors according to the reviewer´s requirements. The authors have filled their contribution in the discussion and conclusion.

The conclusions section also did not respond to the journal's requirements, and it also contains new facts that were not addressed in the article itself (eg COVID-19). There is also a lack of clear guidelines (directions) for possible further research.

This part has been changed by the authors according to the reviewer´s requirements. A lack of clear guidelines (directions) for possible further research has been added in the end of the paper (conclusion).

Reviewer 3 Report

The paper is very well made and the methodology is consistent, although it is not very innovative: the paper is rather an application of existing knowledges than the development of new ones.

The paper is very good so only a few minor changes are required: I would suggest to move sub-section 3.1. and 3.2 to section 2 "Literature review", as it is more appropriate than section 3 "Materials and method".

Author Response

Reference to Review 3:

We would like to thank the reviewer for his insight into the subject addressed.

The paper is very well made and the methodology is consistent, although it is not very innovative: the paper is rather an application of existing knowledge than the development of new ones.

Thank you for your opinion. We  have tried to improve the content of this paper according to different reviewers and we also hope that thanks to the suggestions of reviewers, this paper will be more focused on the scientific research.     

The paper is very good so only a few minor changes are required: I would suggest to move sub-section 3.1. and 3.2 to section 2 "Literature review", as it is more appropriate than section 3 "Materials and method".

We have moved these chapters into the chapter "Literature review" according to your suggestion

Reviewer 4 Report

Please  check the use od english terms also maybe to improve your article you can find folowing article: Vukic, L.: Model of determining the optimal, green transport riute among alternatives: data envelopment// Journal of marine sciences and engineering, 2020., 8.

Author Response

Reference to Review 4:

We would like to thank the reviewer for his insight into the subject addressed.

Please  check the use od english terms also maybe to improve your article you can find folowing article: Vukic, L.: Model of determining the optimal, green transport riute among alternatives: data envelopment// Journal of marine sciences and engineering, 2020., 8.

We have checked the grammar in the paper. Suggested source was also added. 

Round 2

Reviewer 2 Report

Well done. However, I believe that one technical change should be made to the article in order to make the text structurally and logically sound. You wrote "On the other hand we are afraid that the part „Materials and methods“ is not adequate or too short and this change can affect the meaning of this part of the paper." In the light of my previous remark and your insight, I would suggest moving the current sections 2.4, 2.5 and 2.6 to the Methods and Materials section, as this is a description of the methodology on which you present the results and the discussion.

The quality of Figure 2 is poor (maybe because it is in pdf format).

Author Response

We would like to thank the reviewer for his insight into the subject addressed.

Well done. However, I believe that one technical change should be made to the article in order to make the text structurally and logically sound. You wrote "On the other hand we are afraid that the part „Materials and methods „is not adequate or too short and this change can affect the meaning of this part of the paper." In the light of my previous remark and your insight, I would suggest moving the current sections 2.4, 2.5 and 2.6 to the Methods and Materials section, as this is a description of the methodology on which you present the results and the discussion.

We have removed these sections 2.4, 2.5 and 2.6 to the Materials and Methods section. Now they are changed to subchapters 3.1, 3.2 and 3.3 Thank you that you have practically accepted our idea about this issue. We believe that this structure can contribute to better understanding of methods that we have used in our paper.

The quality of Figure 2 is poor (maybe because it is in pdf format)

We have changed the brightness and contrast of this figure. Authors hope that now it is more visible (not only in doc. format but also in pdf format).   
